# Antioxidant Films from Cassava Starch/Gelatin Biocomposite Fortified with Quercetin and TBHQ and Their Applications in Food Models

**DOI:** 10.3390/polym13071117

**Published:** 2021-04-01

**Authors:** Wirongrong Tongdeesoontorn, Lisa J. Mauer, Sasitorn Wongruong, Pensiri Sriburi, Alissara Reungsang, Pornchai Rachtanapun

**Affiliations:** 1School of Agro-Industry, Mae Fah Luang University, Chiang Rai 57100, Thailand; wirongrong.ton@mfu.ac.th; 2Research Group of Innovative Food Packaging and Biomaterials Unit, Mae Fah Luang University, Chiang Rai 57100, Thailand; 3Department of Food Science, Purdue University, West Lafayette, IN 47907, USA; mauerl@purdue.edu; 4Division of Biotechnology, Faculty of Agro-Industry, Chiang Mai University, Chiang Mai 50100, Thailand; sasitorn.w@cmu.ac.th; 5Department of Chemistry, Faculty of Science, Chiang Mai University, Chiang Mai 50200, Thailand; pensiri@scinece.cmu.ac.th; 6Department of Biotechnology, Faculty of Technology, Khon Kaen University, Khon Kaen 40002, Thailand; alissara@kku.ac.th; 7Research Group for Development of Microbial Hydrogen Production Process, Khon Kaen University, Khon Kaen 40002, Thailand; 8Academy of Science, Royal Society of Thailand, Bangkok 10300, Thailand; 9Division of Packaging Technology, Faculty of Agro-Industry, School of Agro-Industry, Chiang Mai University, Chiang Mai 50100, Thailand; 10The Cluster of Agro Bio-Circular-Green Industry (Agro BCG), Chiang Mai University, Chiang Mai 50100, Thailand; 11Center of Excellence in Materials Science and Technology, Chiang Mai University, Chiang Mai 50200, Thailand

**Keywords:** cassava starch/gelatin film, antioxidants, mechanical properties, food, shelf life extension

## Abstract

Edible and active packaging are attractive for use in food packaging applications due to their functionality and sustainability. This research developed new antioxidant active food packaging materials from cassava starch/gelatin (7:3 *w/w*) composite films with varied antioxidant types (quercetin and tertiary butylhydroquinone (TBHQ)) and concentrations (0–200 mg/200 mL film-forming solution) and evaluated their properties. Antioxidant addition altered the mechanical and barrier properties of the films. At 34% relative humidity (RH), increasing the concentration of quercetin increased the tensile strength and decreased the elongation at break of the composite films. Increasing quercetin and TBHQ contents increased the film water solubility and water vapor transmission rate. Intermolecular interactions between the antioxidants and films, as found in Fourier transform infrared (FT-IR) spectra and XRD micrographs, were related to the changed film functionalities. In food application studies, the cassava starch/gelatin films containing quercetin and TBHQ retarded the oxidation of lard (more than 35 days) and delayed the redness discoloration of pork. Cassava starch/gelatin composite films integrated with quercetin and TBHQ can be utilized as active packaging that delays oxidation in foods.

## 1. Introduction

The improvement of active biodegradable films has been increasing because of their potential substitution for some petrochemicals in the food packaging industry and their perceived sustainable choice. Biodegradable films are normally produced from agricultural wastes that contain abundant biopolymeric materials such as polysaccharides, proteins, lipids, or the combination of these components. In efforts to improve functionality, films containing combinations of polymers have been investigated, including those based on polysaccharide and protein blends such as methylcellulose–wheat gluten [1], soluble starch–gelatin [2], hydroxypropyl starch–gelatin [3], soluble starch–caseinate [4], glucomannan–gelatin [5], as well as films based on polysaccharides–polysaccharides mixtures such as starch–methylcellulose [6], pullulan–starch [7], chitosan–starch and chitosan–pullulan [8], CMC–rice starch [9] were studied. Combinations of some polymers may improve the mechanical properties and/or gas/moisture barrier properties of the films, depending on interactions between components within the film structures.

As food additives, antioxidants are commonly used to improve the oxidation stability of lipids and to extend the shelf life, especially for fatty products and foods sensitive to O_2_. Owing to their chemical stability, low cost, and availability, synthetic antioxidants such as butylated hydroxytoluene (BHT), butylated hydroxyanisole (BHA), and tertiary butylhydroquinone (TBHQ) are widely used in the food industry [10]. In the United States, BHA and BHT are approved for human consumption with maximum concentration allowed at 200 mg/kg in fat [11]. TBHQ is considered as the most used antioxidant in vegetable oils [12] because it imparts greater stability than BHT and is more cost-effective [13]. However, their safety remains unclear, and regulations vary by country. There is increasing interest in natural antioxidants (such as ascorbic acid, essential oils, and quercetin) among food manufacturers to serve as substitutes for the synthetic antioxidants [14]. Many natural antioxidants are generally recognized as safe when used in accordance with good manufacturing practices and have added quality and consumer acceptance perceptions attributed to them [11].

Recently, antioxidants have been incorporated into polymer films to advantageously utilize their antioxidative properties in both the films and products to which they are applied [15]. Hargens et al. [16] used tocopherols in edible films to prevent warmed-over flavor and improve precooked meat quality. Oussalah et al. [17] added essential oils into milk protein-based films. Lopez-de-Dicastillo et al. [18] applied catechin and quercetin as natural antioxidants into EVOH-based films. Hanani et al. [19] reported that fish gelatin containing pomegranate (*Punica granatum* L.) peel powder has great potential as an active film with antioxidant and antimicrobial properties, and thus, it can help maintain the quality and prolong the shelf life of food products. Shao et al. [20] added tea polyphenol (TP) into corn distarch phosphate (CDP)/carboxymethyl cellulose (CMC) composite films that showed excellent antibacterial activity against *Staphylococcus aureus*, *Bacillus subtilis*, and *Escherichia coli*. Fang et al. [21] produced starch-based composite films incorporated with salicylic acid (SA) to which the addition of waxy maize starch nanoparticles/κ-carrageenan (WMSNs/KC) significantly increased the tensile strength, water vapor barrier, and thermal stability, while the transparency and elongation at break decreased slightly. Nilsuwan et al. [22] added epigallocatechin gallate to enhance the mechanical and UV-light barrier properties of the fish gelatin-based films. Pellá et al. [23] found that coating guava fruit with cassava starch, casein, and gelatin blends increased its shelf life by two days due to the low water vapor transmission rate of the films, which decreased the fruits’ mass loss. Kowalczyk et al. [24] developed edible films based on pullulan/gelatin blends incorporated with potassium sorbate that increased the pH, UVB-barrier capacity, and moisture absorption properties. 

The cassava starch/gelatin films used as a starting point in this study were effectively prepared and characterized in a previous study [25]. Other related studies showed that tensile strength, elongation, and folding endurance of rice flour/cassava starch blended films were not affected by antioxidants (propyl gallate (PG), BHA, and BHT) [26], and that cassava starch/CMC films incorporated with antioxidants delayed oxidation reactions in lard and pork [10]. Building on these previous studies, the objectives of this study were to develop antioxidant films from cassava starch/gelatin polymer mixtures, observe the effects of antioxidants (a natural and a synthetic antioxidant: quercetin and TBHQ) on the mechanical and structural properties of these films, and apply the antioxidant-containing films to food simulants to document their effects on oxidation reactions therein. 

## 2. Materials and Methods

### 2.1. Materials

The films were made by cassava starch (Bangkok Inter Food Co., LTD., Bangkok, Thailand). Gelatin and 2,2-diphenyl-1-picryl hydrazyl (DPPH) were acquired from Sigma (Sigma-Aldrich, Darmstadt, Germany) and Fluka (Buchs, Switzerland), respectively. Antioxidants, quercetin, and tertiary butylhydroquinone (TBHQ) were purchased from Alfa Aesar (Haverhill, MA, USA) and Fluka (Buchs, Switzerland), respectively. Magnesium chloride and magnesium nitrate from Ajax Finechem (Taren Point, Australia) were used for controlling environmental RH is storage chambers to 34% and 54% RH, respectively. Folin–Ciocalteu reagent, glacial acetic acid, methanol, chloroform, and glycerol were bought from Merck (Darmstadt, Germany). 

### 2.2. Film Preparation

The process of casting the cassava starch/gelatin film was adapted from methods published by Tongdeesoontorn et al. [10,27]. The film-forming solution of cassava starch (7 g) and gelatin (3 g) in distilled water (200 mL) was prepared with different quercetin and TBHQ contents (0, 20, 50, and 100 mg) (Table 1). Glycerol (30% *w*/*solid w* cassava starch/gelatin) was added as a plasticizer. To achieve starch gelatinization, film solutions were heated to 80 °C with continuous agitation. Then, the film-forming solutions were cast on Teflon plates and dried for 24 h at room temperature (25 °C).

### 2.3. Film Mechanical Properties

Films were cut into 100 × 25.4 mm strips and preconditioned in the environmental chambers at 34 and 54% RH at 25 °C for 48 h [28]. Tensile strength (TS) and film break elongation were determined using a universal testing machine (Hounsfield, UK) according to the ASTM D 882-911 method [29]. The initial grip length was 50 mm and pulling speed was 50 mm/min. Twenty specimens were evaluated from each % RH. 

### 2.4. Fourier Transform Infrared Spectroscopy (FT-IR) 

Transmission infrared spectra of the films were documented at 25 °C in the range of 4000–400 cm^−1^ using a Nicolet 6700 Fourier transform infrared (FT-IR) spectrometer (Thermo Electron Corporation, Waltham, MA, USA) following the method of Tongdeesoontorn et al. [10]. 

### 2.5. Differential Scanning Calorimetry (DSC)

Differential scanning calorimetry analyses were done as described by Tongdeesoontorn et al. [10] by using a Mettler Toledo Schwerzenbach instrument (Polaris Parkway Columbus, OH, USA). Three replicates of film samples (≈10 milligrams) in sealed aluminum pans were heated in the temperature range −20 to 220 °C at a heating rate of 5 °C/min in a nitrogen atmosphere (50 mL/min). 

### 2.6. X-ray Diffraction (XRD) 

The X-ray diffraction patterns of cassava starch/gelatin films with and without antioxidants were collected using an X’Pert MPD X-ray diffractometer (Philips, The Netherlands) at 40 kV and 35 mA in the 2θ range of 5–50°. 

### 2.7. Film Morphology

Film samples were preconditioned at 25 °C, 54% RH for 7 days prior to analysis. The surface and cross-section microstructure of the films were observed by a Scanning Electron Microscope (JEOL JSM-6700F, Tokyo, Japan) with gold sputtering. 

### 2.8. Water Solubility of Cassava Starch/Gelatin Films

Film solubility was investigated as a percentage of solubilized film in water following a method of Tongdeesoontorn et al. [10]. Films were dried at 65 °C for 24 h followed by placement in a desiccator with dried silica gel for 2 days. The initial dry weights (*w_i_*) of films (about 0.3 g) were checked, and then films were added into beakers with 50 mL of distilled water at 25 °C and agitated intermittently for 24 h. The solutions comprising film residues were filtered with dried Whatman filtered paper No.1. The residues on filtered papers were dried at 80 °C for 24 h and then weighed to determine the final dry weight (*w_f_*). Tests were achieved in triplicate, and the water solubility (%) was calculated using Equation (1):(1)$$\mathrm{Water}\;\mathrm{solubility}\left(\%\right)\;=\;\left(\frac{w_{i}\;-\;w_{f}}{w_{i}}\right)\times 100.$$

### 2.9. Water Vapor Transmission Rate (WVTR)

The water vapor transmission rate (*WVTR*) of films was determined at 25 ± 1 °C using the ASTM E96-80 method [30]. Ten grams of dried silica gel were placed inside the aluminum cup, and then, a circular area (0.00328 m^2^) of the cup was sealed over with each film sample. Cups were stored in desiccators containing saturated salt solution (MgCl_2_ and Mg(NO_3_)_2_ for 34% and 54% RH, respectively). The *WVTR* was calculated using the following Equation (2):(2)$$WVTR\;=\;\frac{\Delta W\;\times \;A}{\Delta t}\;(g/m^{2}\;\mathrm{day}.)$$
where ∆*W/*∆*t* is the amount of water gain (g/day) and *A* is the test area (m^2^).

### 2.10. Total Phenolic Content Assay

The Folin–Ciocalteu phenol reagent was used to obtain the concentration of total phenolic compound content in the cassava starch/gelatin films with antioxidants [31]. For this purpose, 2 × 2 cm (~0.1 g) of each film sample was liquefied in 10 mL of methanol for 24 h to extract phenolic compounds from the film specimens [10]. According to the Folin–Ciocalteu method of Association of Official Analytical Chemists (AOAC) [32] with slight modifications, the total phenolic content (TPC) of the composite film samples was then determined. Briefly, methanolic film extracts (0.5 mL) were mixed with 8 mL of distilled water and 1 mL of Folin–Ciocalteu reagent. The blend was gestated for 5 min before adding 0.5 mL of saturated sodium carbonate and then stored in the dark for 30 min. The absorbance at 760 nm of the mixture was analyzed using a spectrophotometer (SPECTRO SC LABOMED, Los Angeles, CA, USA), and methanol was used as a blank. The total phenolic compound concentration in the samples was expressed as the amount of gallic acid equivalent as described by Siripatrawan et al. [33] and using Equation (3):(3)$$A_{760}=0.001\;\mathrm{mg}\;gallic\;acid\;+\;0.027$$
where *A*_760_ is the absorbance at 760 nm.

### 2.11. Antioxidant Activity Determination in the Composite Films 

A free radical scavenging assay using 2,2-diphenyl-1-picryl hydrazyl (DPPH) was used to compare the short-term antioxidant potencies of antioxidants in the cassava starch/gelatin films [10,34]. Film samples (2 × 2 cm) were liquefied in 10 mL methanol for 24 h and filtered. The filtrate (500 μL) was combined with 2 mL of 0.06 mM DPPH solution (in methanol) and then incubated for 30 min at room temperature in a dark location. The absorbance at 517 nm was measured with a spectrophotometer (SPECTRO SC LABOMED, Los Angeles, CA, USA). Methanol and quercetin were used as a reference and positive control, respectively. The DPPH radical scavenging activity was calculated according to the following Equation (4): (4)$$\%Radical\;scavenging\;activity\;=\left(\frac{A_{reference}-A_{sample}}{A_{sample}}\right)\;\times \;100$$
where *A* is the absorbance at 517 nm.

### 2.12. Application of Cassava Starch/Gelatin Film Incorporated with Antioxidant

#### 2.12.1. Effect of Antioxidant Addition into Cassava Starch/Gelatin Films on Lard Rancidity during Storage 

The effects of antioxidant (quercetin and TBHQ) addition to films on the quality changes of lard during storage were studied using methods adapted from previous publications [10,35]. Lard samples (15 mL) were packed in cassava starch/gelatin-antioxidant (quercetin or TBHQ) heat-sealed bags with film areas of 100 cm^2^ and kept at 30 °C, 40%RH for 35 days. The control was put in beaker without any package. The peroxide value of lard was analyzed at 1-week intervals.

#### 2.12.2. Estimation of Peroxide Value (PV)

The peroxide value (PV) measures primary products of lipid oxidation and is used to determine the oxidative state of lipid-containing foods [36]. Lard was extracted with a mixture of water, methanol, and chloroform (30:50:100), and PVs of the extracts were investigated using the modified method of Jung et al. [36]. The extract (1 g) was dissolved in 25 mL of solvent (chloroform/acetic acid as 2: 3 *v/v*). Then, saturated potassium iodide (1 mL) was added, and the solution was retained in the dark for 10 min. After stabilization, 30 mL of distilled water was added into the solution, which was followed with 1 mL of starch solution (1% *w/v*), and solutions were then titrated with 0.01 N Na_2_S_2_O_3_ until colorless. PVs were calculated as follows (Equation (5)):(5)$$Peroxide\;Value\;\left(PV\right)\;=\frac{\left(S-B\right)\times N\times 1000}{W}$$
where *S*, *B*, *N* (mol equiv/L), and *W* denote the titration quantity of sample, the titration quantity of blank, the normality of Na_2_S_2_O_3_, and the sample weight (*W*, g), respectively [10].

### 2.13. Antioxidant Films Application on Fresh Pork 

Antioxidant film applications on fresh pork were studied [10,17,36]. Fresh pork samples were obtained from a butcher shop, sliced into 5 × 10 × 1.5 cm (width × length × thickness) sections, and weighed ca. 30–35 g. A piece of sliced pork was placed on a polystyrene tray (10 × 5 × 1.5 cm) and covered on the top side with one sheet of the antioxidant cassava starch/gelatin composite films. Then, trays were covered with polyvinylchloride (PVC) film and kept at 4 ± 1 °C. During storage, the color of the pork was checked periodically (days 0, 4, 8, and 12) using a hand-held colorimeter (Minolta, Tokyo, Japan) to document the a* values (positive value = redness) of the fresh pork. Percentage of pork redness reduction was calculated from a* value following Equation (6):(6)$$\%\;\mathrm{Redness}\;\mathrm{decrease}\;=\frac{{(a}_{0}^{*}{-\;a}_{t}^{*})\times \;\;100}{a_{0}^{*}}\;$$
where $a_{0}^{*}$ is the a* value of sample at day 0, and $a_{t}^{*}$ is the a* value at the storage time.

### 2.14. Statistical Analysis

All results were analyzed statistically by ANOVA analysis and Duncan’s multiple range tests to determine the significant differences between group samples using a statistical program SPSS v. 10.0 at a confidence interval of 95%.

## 3. Results and Discussion

### 3.1. Mechanical Properties

Cassava starch/gelatin (7:3) films are reported to have good mechanical properties [27]. To determine the effects of antioxidants on mechanical properties, quercetin and TBHQ were added to the cassava starch/gelatin films. The tensile strengths (TS) of cassava starch/gelatin films containing the different concentrations of quercetin and TBHQ after equilibration at two RHs are shown in Figure 1a.

Increasing storage RH decreased the TS of all films, as expected, since water plasticizes films. Adding antioxidants to the films had type and concentration-dependent effects on TS. The cassava starch/gelatin films with lower amounts of quercetin showed higher TS than the control at 34%RH, which is attributed to the intermolecular hydrogen bonding interactions formed between hydroxyl groups of quercetin (hydrogen bond donors) and the starch/gelatin molecules. Intermolecular hydrogen bond networks can increase the strength of films [37]. Increasing the quercetin concentration to 200 mg and increasing the storage RH both decreased the TS of the films compared to the control. In contrast, films containing TBHQ had lower TS than the control at all TBHQ concentrations, which was attributed to the possible incompatibility of the TBHQ and the starch/gelatin polymers, which potentially disrupted the polymer network, thereby decreasing the film strength. This result is consistent with reports of TS of fish skin gelatin film incorporated with BHT and α-tocopherol [34] and TS of chitosan film with α-tocopherol [38]. For films stored at the higher 54%RH, the addition of either antioxidant at any concentration decreased TS of the films, indicating that the antioxidants likely altered both film network structure and interaction with water. 

The elongation at break (EAB) traits of the films are shown in Figure 1b. Increasing storage RH increased the EAB of all films. Antioxidant type and concentration had variable effects on the EAB. Adding TBHQ to the films increased EAB compared to the control; however, TBHQ concentration was not a significant factor. Conversely, films with lower concentrations of quercetin had lower EAB than control films at 34%RH but higher EAB than the controls at 54%RH. Increasing the quercetin concentration to 200 mg increased EAB at 34%RH to slightly above that of the control. At 54%RH, quercetin-containing films had higher EAV than the control, but varying quercetin concentration did not significantly affect EAB.

By comparing the mechanical properties of quercetin and TBHQ films, it is evident that cassava starch/gelatin composite films containing quercetin had a higher tensile strength than TBHQ-containing films. Additionally, cassava starch/gelatin films with quercetin were less flexible than films with TBHQ. The composite films with the highest TS were films made from 50 mg quercetin/200 mL film solutions and stored at 34%RH, while the highest EAB films were made from 100 mg TBHQ/200 mL film solutions and stored at 54%RH. These findings were not consistent with a report by Rachtanapun et al. [26], who studied the addition of antioxidants (PG, BHA, BHT) to rice flour/cassava starch films and found that the type of antioxidants had no effect on the mechanical properties of the film [26]. In terms of the storage RH effects on the film mechanical properties, the EAB of films stored at 54%RH was greater than those stored at 34%RH, but the TS were lower. Increasing the storage RH reduced intermolecular interactions and increased polymer mobility, both of which result from water acting as a plasticizer by binding with hydroxyl groups (OH) in the starch chain [39,40]. Others have reported similar RH-dependent effects on film mechanical properties [41]. 

### 3.2. Fourier Transform Infrared Spectroscopy (FT-IR) 

FT-IR spectroscopy was used as a tool to investigate the structure and possible interactions between cassava starch/gelatin films and antioxidants (quercetin and TBHQ). Special note should be made of the peaks between 3272 and 2930 cm^−1^ (Figure 2 and Table 2), referring to the expansion vibration of the free hydroxyl groups and the asymmetrical and symmetrical extension of the N–H bonds in the amino groups, respectively [33]. In contrast to films containing antioxidants, these peaks are more prominent in the control films (Table 1). In addition, a strong band at 1407 cm^−1^, associated with OH in-plane bending [33], is discernible in the films with antioxidants. There are also important differences at 1650, 1550, and 1000 cm^−1^. The peaks at 1650 cm^−1^ are ascribable to carbon-to-oxygen (C=O) stretching within the carboxylic group (amide I), and the peak at 1550 cm^−1^ is correlated to the N–H bond (amide II) stretching of gelatin [34]. CH–O–CH_2_ stretching is causing the band at 1076 cm^−1^ [42]. Adding antioxidants into cassava starch/gelatin films resulted in shifting the O–H and –N–H bands to 3278–3272 cm^−1^ and 2929–2926 cm^−1^, respectively. With antioxidant addition, the band of C–OH bending of the cassava starch/gelatin film that appeared at 1322 cm^−1^ was displaced to 1326–1323 cm^−1^. With the addition of quercetin and TBHQ, minor differences in the absorption band strength at 1650 and 1550 cm^−1^ were observed in the composite films. These findings were consistent with the FT-IR spectra of fish gelatin films containing BHT and α-tocopherol [34]. Increasing amide I and amide II absorbance implied that the adding of quercetin and TBHQ may affect the gelatin structure [34].

A rise in the absorption bands in the FT-IR range at 1650 and 1550 cm^−1^ correlated with a decrease in peaks at 3273–2930 cm^−1^. This discovery confirmed the theory that because of the intermolecular interactions between antioxidant polyphenol compounds with hydroxyl and amino groups of the gelatin matrix, there may be a clear structural change in the films [33,43]. Moreover, these results also corresponded with the FT-IR spectra of chitosan films incorporated with green tea extract [33] and the study of physicochemical interactions between chitosan and catechin by Zhang and Kosaraju [44], who found that when catechin was added, the amine functional groups of the chitosan decreased. Similar results in the formation of covalent bonds between gallic acid–chitosan and catechin–chitosan in films have also been observed by Curcio et al. [31]. 

It is apparent from the FT-IR analyses that the incorporation of quercetin and TBHQ could form hydrogen bonding and thus occupy the cassava starch/gelatin matrix functional group, consequently reducing the free hydrogen group that can form hydrophilic bonding with water [33]. These structural differences resulted in the differences in mechanical properties of the films shown in Figure 1.

### 3.3. Thermal Properties

DSC thermograms of the films are provided in Figure 3, and the melting temperature (Tm) and heat of fusion (ΔH) of cassava starch/gelatin films containing different concentrations of quercetin and TBHQ are shown in Table 3. DSC thermograms of corn starch/gelatin films [45] and soluble starch/gelatin films [2] also show similar results to the control films. Adding either of the antioxidants to the cassava starch/gelatin films resulted in an increase in the Tm of the films to higher than that of the control film and either antioxidant ingredient alone, and the ΔH of the antioxidant-containing films was higher than that of the control filoms. The sharp endothermic peak in the thermograms of films containing both antioxidants (Figure 3) is an indication of homogeneity and has been related to the melting of crystalline starch and gelatin domains [46]. The Tms of films containing TBHQ were higher than Tms of films containing quercetin, which was attributed to the higher Tm of TBHQ than quercetin. The higher ΔH in the films containing antioxidants is indicative that more energy is needed to break the bonds of interaction between quercetin or TBHQ and the film matrix [47,48]. 

### 3.4. X-ray Diffraction Patterns

The X-ray diffraction patterns of the control films and antioxidants, along with those of the antioxidant-containing films, are shown in Figure 4. The diffractograms were obtained by following the methods of Almasi H. et al. [45]. The diffractogram pattern of gelatin has a characteristic sharp peak at 2θ = 28° and a peak in the region of 2θ = 7–8° [45]. The control films had a characterisic sharp peak in the diffractorgrams at 2θ = 28° and an amorphous pattern from 2θ = 15–25°. The individual antioxidants had distinct peak patterns associated with their crystallinity, as shown in Figure 4. Adding quercetin to the films resulted in a decrease in the intensity of the peak located at 2θ = 28°, and at the higher concentrations of quercetin addition, a peak began to appear near 12°, which was one of the more prominent peaks in the quercetin diffractogram. Adding TBHQ to the films also decreased the intesntiy of the peak at 28°.

The decreased peak intensity in the films at 28° is related to the re-arrangement and possible blockage of starch/gelatin crystallization [49,50]. A combination of the low amount of antioxidant addition and their intermolecular interactions with the polymer film matrix resulted in no characteristic peaks in the diffractorgrams associated with antioxidant crystallinity.

### 3.5. Film Morphology

The scanning electron micrographs of the cross-sections of composite films at different concentration of quercetin and TBHQ are presented in Figure 5. 

All composite films showed similar surface images to the microstructure of cassava starch-gum bio-based films [51], which are visibly smooth and compact. 

The SEM image of cassava starch/gelatin film showed a dense structure with a few areas of roughness (Figure 5a), which can happen due to the versatility of gelatin films [52]), which proved a homogeneous structure. The micrographs of cassava starch/gelatin films with quercetin and TBHQ (Figure 5b,c, respectively) exhibited a dense and rough cross-section. More cracking was seen in the SEM micrograph of the TBHQ-containing films, which might have been caused by phase separation or inhomogeneity of TBHQ in the polymer matrix and could have contributed to the lower tensile strength of the TBHQ-containing films (Figure 1).

### 3.6. Water Solubility 

Water solubility is an important trait of many composite films. The water solubility can be essential for some applications in the food industry, such as food or additives encapsulation [48], while for other applications, the insolubility of films in water is important (including moisture barrier applications). Adding TBHQ and quercetin to cassava starch/gelatin composite films increased the water solubility of the films compared to the controls (Figure 6). The water solubility of the composite films tended to increase along with the concentration of quercetin and TBHQ—the least soluble films were the controls (about 45%), and the most soluble films contained 200 mg TBHQ (about 65% soluble). Phenolic compounds and compounds and flavonoids have polar groups (–OH) [49] and thus are expected to interact with water. Similar results were found for the solubility of rice flour/cassava starch films upon the addition of gallic acid [50].

### 3.7. Water Vapor Transmission Rate (WVTR) 

Water vapor transmission rates (WVTRs) of cassava starch/gelatin films with and without TBHQ and quercetin are shown in Figure 7. WVTR increased in all films from 34% to 54% RH. At a given storage RH and concentration, films containing quercetin tended to have higher WVTR than films containing TBHQ. Films containing antioxidants had drastically higher WVTR than control films, regardless of antioxidant type or concentration, with the largest increases in WVTR found at the highest storage RH. Water is a plasticizer, increasing polymer chain mobility and reducing intermolecular forces in film structures as the amount of water increases [51]. Therefore, increasing storage RH is expected to increase WVTR. Both the physical film structure (e.g., the polymer matrix) and the solubility of water in the matrix (or the matrix in water) will affect WVTR. The mechanical properties and spectra of the films were different between the different antioxidant formulations (Figure 1 and Table 1), indicating that the film matrices were different and would therefore have different diffusion pathways for water to move through the films upon antioxidant addition. Additionally, quercetin has more hydroxyl groups and is more polar than TBHQ, which also seem to be contributing factors that result in the higher WVTR of the quercetin-containing films. Others have found similar increases in WVTR when antioxidants are added to polymer films. Adding PG, GA, and BHA to rice flour–cassava starch films increased WVTR at 25 °C, 45%RH [52]. Adding oregano extract increased the water vapor permeabilities (WVPs) of soy protein films [49]. Oregano extracts contain polar compounds that increased the hydrophilic properties of the films and concomitantly increased WVP values [53]. The permeability of starch films can depend on many factors, such as the ratio of amylose to amylopectin, polymer packing, chemical structure, plasticizer, crystallinity, and environmental humidity [54]. The WVTR was affected by the structure of the polymers inside the film matrix [55]. The presence of the antioxidants decreased the crystallinity of the cassava starch/gelatin films (Figure 4), which would have changed the diffusion pathways and increased permeability.

### 3.8. Total Phenolic Assay

As the concentration of antioxidants increased, the total phenolic content of cassava starch/gelatin films also increased significantly (Figure 8). This result agreed with the increase of total phenolic content in chitosan films with increasing green tea extract [33]. The total phenolic content of cassava starch/gelatin films with quercetin was greater than that of films with TBHQ at the same antioxidant concentration, since quercetin has a higher molecular weight than TBHQ. The total phenolic contents of quercetin and TBHQ films were stable throughout the storage period (30 days), indicating that quercetin and TBHQ could be retained in cassava starch/gelatin films for a long duration.

### 3.9. Antioxidant Activity of the Composite Films 

The results displayed that increasing quercetin and TBHQ concentration had no effect (*p* ≥ 0.05) on the DPPH scavenging activity of cassava starch/gelatin films (data not shown), which represented the similar antioxidant activity of all cassava starch/gelatin films with different antioxidant contents. The DPPH radical scavenging activity of the cassava starch/gelatin films were not affected by the storage time.

### 3.10. Application of Antioxidant Films on Lard

The lipid peroxide value (PV) was determined for lard samples stored in cassava starch/gelatin films containing quercetin or TBHQ during storage to document the film effects on lipid oxidation (Figure 9). After a 15-day storage time, the PV of the unpackaged lard (control) increased from 5 to >15 meq/kg, which suggested rancidity. No significant increase in PV was found in lard samples stored in quercentin or TBHQ-containing films after 15 days of storage. Between days 15 and 35 of storage, the PV of lard stored in the antioxidant-containing films did increase, but it remained below 20 meq/kg for a 35-day storage period. This result demonstrated that packaging with cassava starch/gelatin films containing quercetin and TBHQ could delay the oxidation of lard. Compared with the control, the PV increase of lard packaged in cassava starch gelatin films containing quercetin and TBHQ was significantly lower, implying that a continued release of quercetin and TBHQ from the cassava/gelatin films was likely to delay lipid oxidation [36]. This result agreed with the delayed oxidation of lard packaged in fish skin gelatin film with BHT and α-tocopherol [34], in soy protein isolate film with BHA and blueberry extract [35], in rice flour/cassava starch film containing PG, BHA, and BHT [26], and in cassava starch–CMC films combined with quercetin and TBHQ [10].

### 3.11. Application of Antioxidant Films on Fresh Pork 

Meat surface discoloration largely depends on the oxidation rate of the red oxymyoglobin into metmyoglobin, which gives the meat less attractive brown color [56]. The effectiveness of antioxidants for inhibiting lipid peroxidation and color change in minced meat products has been well documented [17,36,56,57]. 

In this research, pork samples were coated with cassava starch/gelatin films containing different concentrations of quercetin and TBHQ, and color of pork was then monitored for 12 days. The percent reduction in redness was measured, as shown in Figure 10, in order to compare film effects on the reduction in redness. After 8 days of storage, the pork samples in contact with films containing quercetin and TBHQ had more (*p* < 0.05) redness than the control (uncovered pork). The redness of uncovered pork reduced rapidly and decreased by more than 50% after 8 days of storage, while the redness of pork covered by quercetin and TBHQ films showed a lower drop of redness than control (<10% for all but one film type). Pork covered with TBHQ films showed a lower redness reduction than quercetin films. After 12 days, the redness of pork concealed with TBHQ films decreased slightly (less than 20% reduction in redness). Increasing the content of quercetin decreased the rate of color change, but variations in TBHQ concentration did not have a major difference in reducing pork redness. These findings suggested that the addition of quercetin and TBHQ in the cassava starch/gelatin film provided good results in the prevention of overall discoloration. Moreover, these findings also corresponded with the inhibition of pork oxidation by anti-oxidative plastic films coated with horseradish extract [36], using tea catechin infused PVA–starch film on red meat [58] and pork covered with cassava starch-CMC films combined with quercetin and TBHQ [10].

The oxidation of myoglobin in pork can be inhibited by the cassava starch/gelatin antioxidant composite films (Figure 10). This result verified the antioxidant activity of cassava starch/gelatin film with quercetin and TBHQ, which could slow down color discoloration and prolong the oxidation of lard of quercetin and TBHQ films. 

## 4. Conclusions

The addition of quercetin and TBHQ as well as environmental relative humidity influenced the mechanical properties of cassava starch/gelatin films. Cassava starch/gelatin films containing quercetin displayed better tensile strength but lesser break elongation than control films. Conversely, films containing TBHQ showed lower tensile strength but higher elongation at break than control films. Changes in polymer matrix structures in the presence of the different antioxidants were documented by FT-IR and XRD analyses. These structural changes corresponded to differences in mechanical and barrier properties of the films. Increasing the content of quercetin and TBHQ increased the film solubility in water and the WVTR of composite films. Throughout the storage period (30 days), both total phenolic content and antioxidant activity stayed in films. For a potential application, cassava starch/gelatin film comprising quercetin and TBHQ were found to slow down the oxidation of lard (more than 35 days) and postpone the redness discoloration of pork. The developed quercetin-containing and TBHQ-containing cassava starch/gelatin films have the ability to be used as active biodegradable films for food products with the potential to delay oxidation reactions in packaged products thereby enhancing product shelf-life and quality.

## Figures and Tables

**Figure 1 polymers-13-01117-f001:**
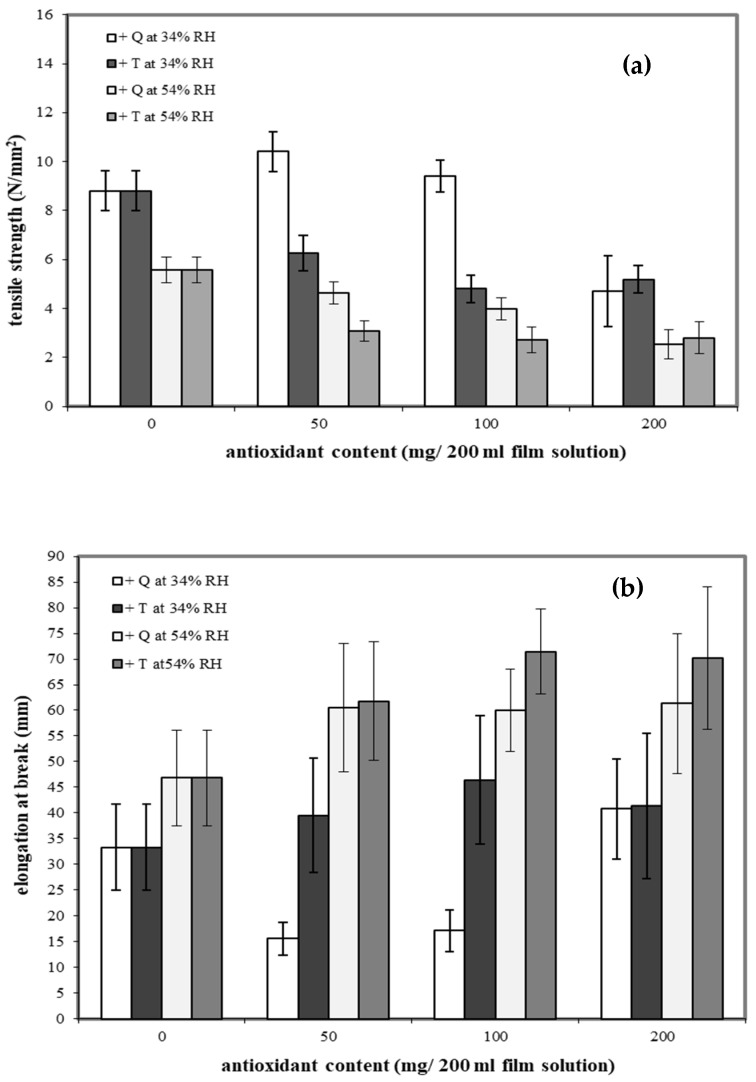
Effect of antioxidant (quercetin (Q) and tertiary butylhydroquinone (TBHQ, or T)) addition and relative humidity on tensile strength (**a**) and elongation at break (**b**) of cassava starch/gelatin composite films.

**Figure 2 polymers-13-01117-f002:**
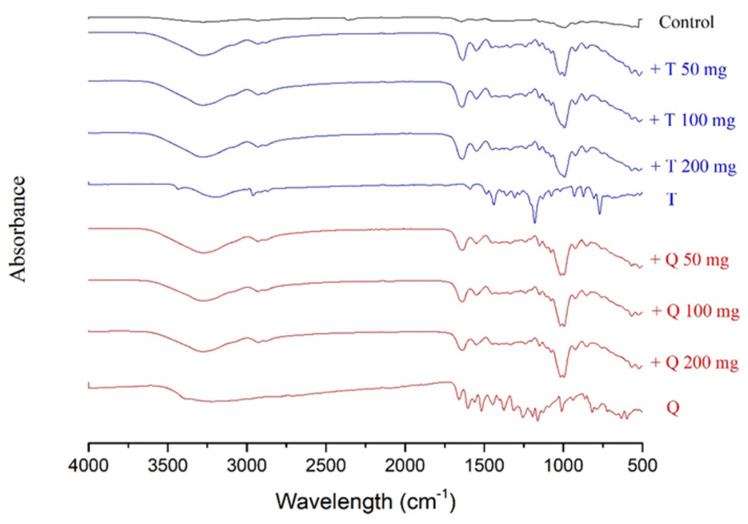
Fourier transform infrared (FT-IR) spectra of the cassava/gelatin films at different concentrations of quercetin (Q) and TBHQ (T) (without antioxidant serves as control).

**Figure 3 polymers-13-01117-f003:**
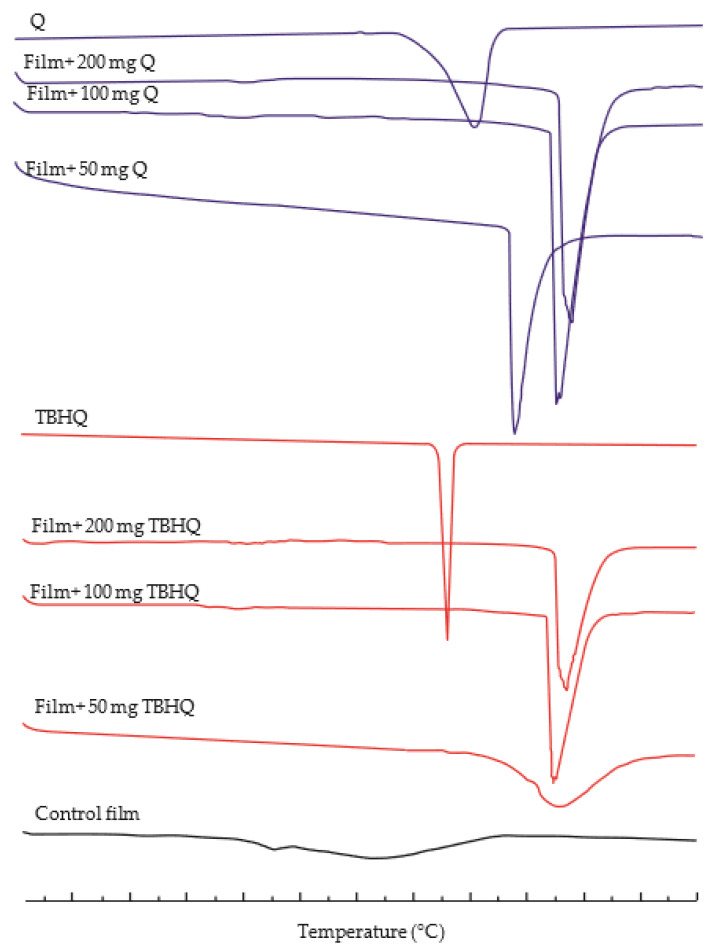
The Differential Scanning Calorimetry (DSC) thermograms of cassava starch/gelatin composite films without antioxidant, with quercetin (Q), and with TBHQ (T).

**Figure 4 polymers-13-01117-f004:**
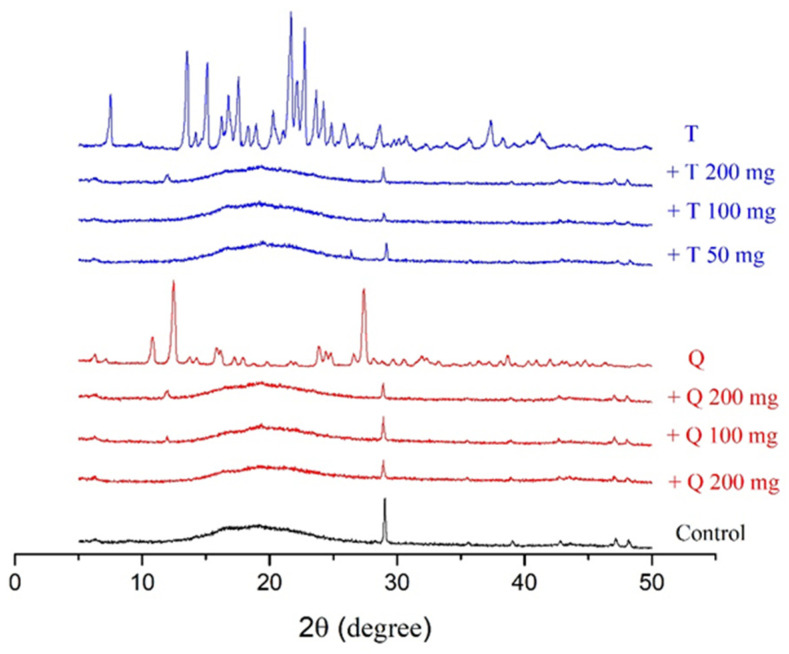
XRD peaks of cassava starch/gelatin films with quercetin (Q) and TBHQ (T).

**Figure 5 polymers-13-01117-f005:**
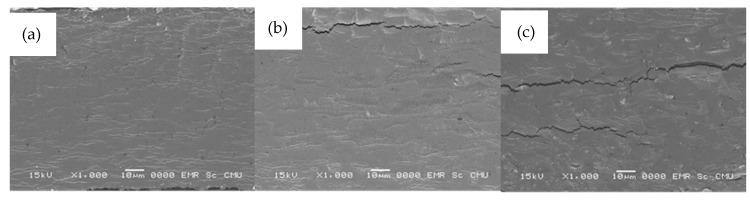
SEM micrographs (×1000) of cassava starch/gelatin composite films (**a**) without antioxidant, (**b**) with quercetin, and (**c**) with TBHQ.

**Figure 6 polymers-13-01117-f006:**
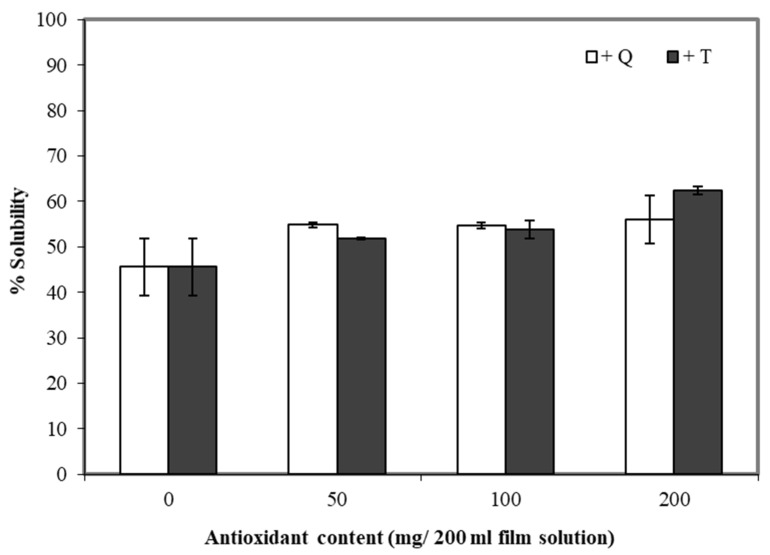
Effect of quercetin (Q) and TBHQ (T) incorporation on water solubility of cassava starch/gelatin films.

**Figure 7 polymers-13-01117-f007:**
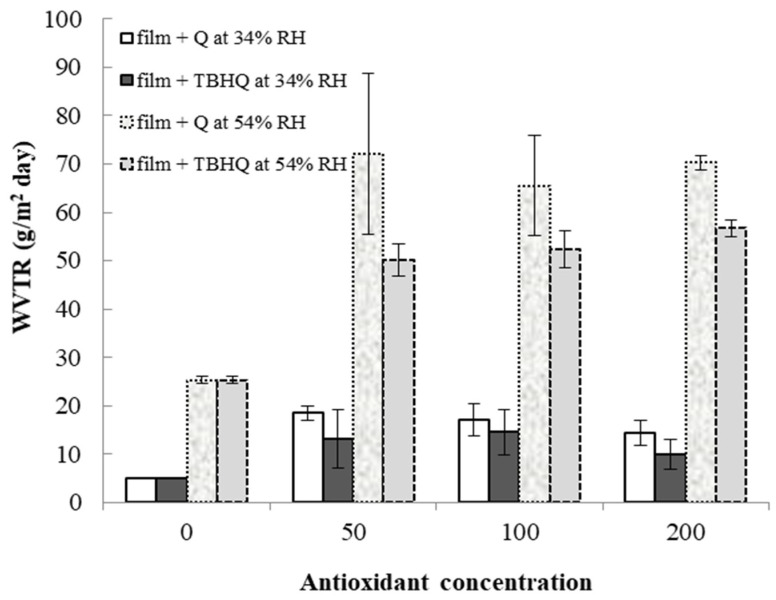
Water vapor transmission rates of cassava starch/gelatin films with quercetin (Q) and TBHQ (T) at 34 and 54% RH.

**Figure 8 polymers-13-01117-f008:**
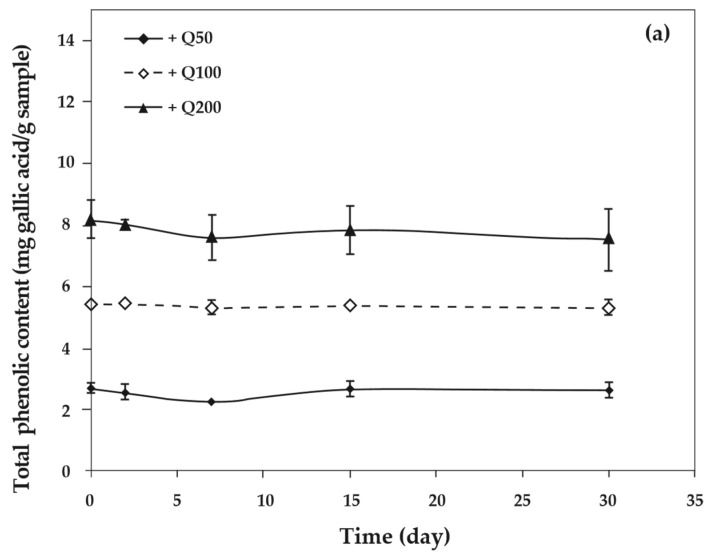
Relationship between storage time and total phenolic content of cassava starch/gelatin films with (**a**) quercetin and (**b**) TBHQ.

**Figure 9 polymers-13-01117-f009:**
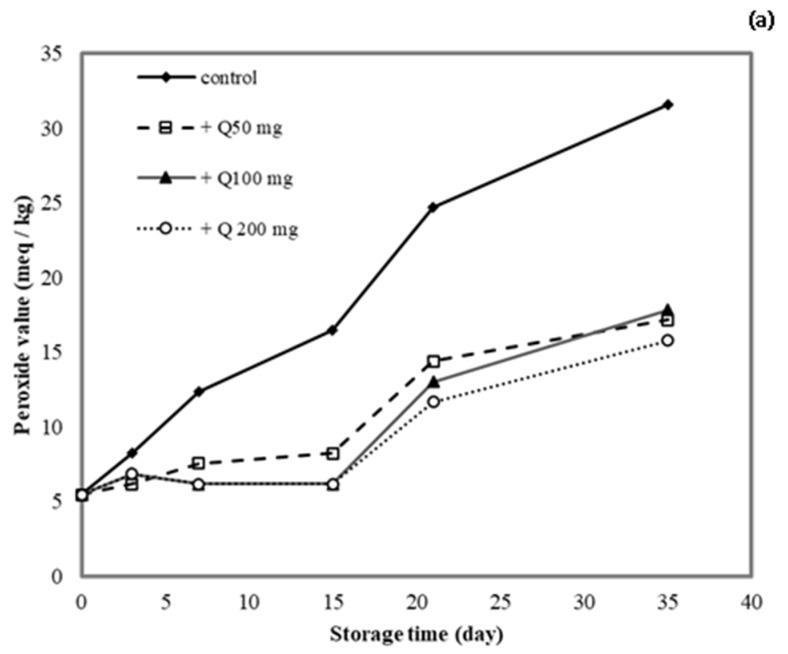
Effect (**a**) quercetin and (**b**) TBHQ addition into cassava starch/gelatin films on the peroxide value of lard.

**Figure 10 polymers-13-01117-f010:**
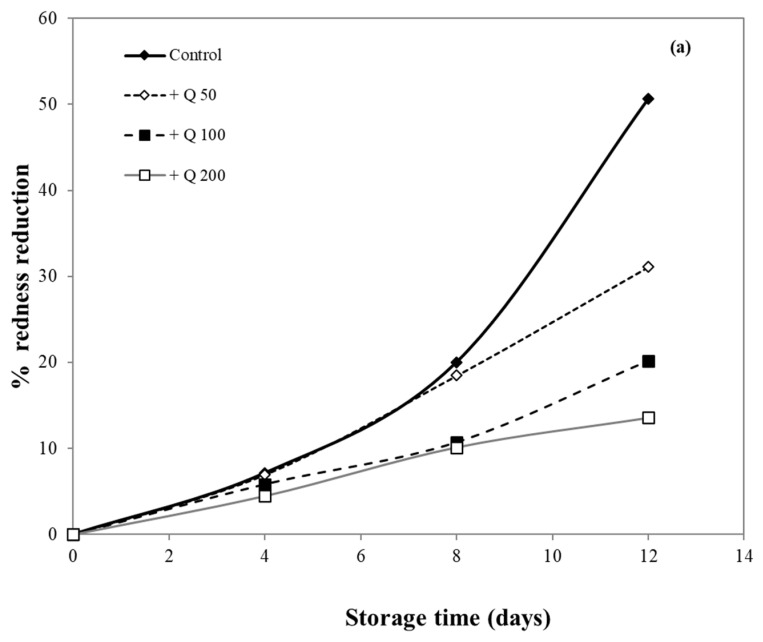
Effect of antioxidant ((**a**) quercetin and (**b**) TBHQ) addition into cassava starch/gelatin film on the percentage redness decrease of pork compared to uncovered (control).

**Table 1 polymers-13-01117-t001:** Composition of film solutions.

Film Samples	Composition of Film Solution (g/200 mL Water)				
	Starch	Gelatin	Glycerol	Quercetin	TBHQ
Control	7	3	3	-	-
Q 50	7	3	3	0.05	-
Q 100	7	3	3	0.10	-
Q 200	7	3	3	0.20	-
T 50	7	3	3	-	0.05
T 100	7	3	3	-	0.10
T 200	7	3	3	-	0.20

**Table 2 polymers-13-01117-t002:** FT-IR peaks wavenumbers of the cassava/gelatin films at different concentrations of quercetin (Q) and TBHQ (T) (without antioxidant serves as control).

Wavenumbers of Peaks (cm^−1^)						
Control Film	Film + Q50	Film + Q100	Film + Q200	Film + T50	Film + T100	Film + T200
3272.9	3272.3	3277.1	3276.5	3274.9	3276.5	3278.3
2930.6	2928.5	2928.3	2926.8	2930.6	2928.8	2929.0
1646.7	1637.3	1636.4	1636.7	1634.6	1638.2	1637.2
1550.0	1550.3	1549.2	1549.6	1550.0	1550.6	1549.8
1407.2	1405.8	1405.1	1405.4	1405.2	1405.6	1405.5
1335.1	1335.1	1335.1	1334.7	1335.9	1335.7	1335.4
1239.9	1239.5	1238.3	1238.7	1238.5	1238.8	1238.5
1149.8	1150.6	1150.6	1150.6	1151.0	1150.5	1150.8
1076.5	1078.4	1078.0	1077.6	1077.8	1077.5	1077.4
-	1017.3	1018.4	1017.6	1020.2	-	-
992.3	996.0	994.4	995.5	992.9	992.6	993.4
923.8	924.3	924.1	924.2	923.4	923.8	923.8
852.6	852.6	851.8	853.1	852.4	854.2	853.8

**Table 3 polymers-13-01117-t003:** Melting temperature (T_m_) and heat of fusion (ΔH) of cassava starch/gelatin film incorporated with quercetin (Q) and TBHQ (T).

Films	T_m_ (°C)	ΔH (J/g)
Cassava starch-gelatin	110.59 ± 2.37 ^a^	71.21 ± 1.99 ^a^
Quercetin	128.88 ± 3.18 ^b^	151.00 ± 4.66 ^b^
TBHQ	131.19 ± 5.70 ^b^	273.56 ± 9.77 ^e^
+ 50 mg Q	155.93 ± 3.92 ^c^	145.09 ± 3.38 ^bc^
+ 100 mg Q	177.50 ± 1.87 ^f^	166.79 ± 1.57 ^bc^
+ 200 mg Q	174.41 ± 0.48	176.42 ± 2.71 ^bcd^
+ 50 mg T	171.34 ± 2.15	190.50 ± 13.83 ^cd^
+ 100 mg T	166.31 ± 3.73 ^d^	201.08 ± 6.73 ^d^
+ 200 mg T	171.68 ± 0.39 ^e^	206.29 ± 10.52 ^d^

Different letters in the same column show significant differences between the means attained in Duncan’s test (*p* < 0.05).

## Data Availability

The data presented in this study are available on request from the corresponding author.

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
