# Peer review of "Antioxidant Films from Cassava Starch/Gelatin Biocomposite Fortified with Quercetin and TBHQ and Their Applications in Food Models"

_polymers, 2021, doi:10.3390/polym13071117_

Round 1
Reviewer 1 Report
This research has developed new antioxidant active food packaging materials from cassava starch/gelatin (7:3 w/w) composite films varied with antioxidant types [quercetin and tertiary butylhydroquinone (TBHQ)] and concentration (0-200 mg / 200 mL film-forming solution) and evaluated their properties. At 34% RH, increasing the concentration of quercetin affects composite films by increasing the tensile strength and reducing the elongation at break. Cassava starch/gelatin composite films integrating with quercetin and TBHQ 40 can be utilized as an active packaging which can delay oxidation in foods.
- “However, there was no effect on EAB of the cassava starch/gelatin films when quercetin concentration is increased.”. The authors should explain this result.
- 3 (XRD) 8 (a). are not very clear. Please improve resolution of the Figs, especially the numbers and words.
- “In addition, a strong water band at 1407 cm-1, associated with OH in-plane bending, is discernible in the films with antioxidants.”. One reference is needed.
- FTIR spectra are recommended to be placed in the text.
- The letter notation of Duncan’s test using ANOVA in Table 2 was all wrong. Please correct the rule based on letter marking.
- “3.5 Film morphology”. The author should discuss the differences between the treatments and explain the potential significance.
- Combine Figures 2a and 2b into one picture. Figure 3 is the same process.
- Many irregularities were described in the manuscript. For example, “14.6 – 21.0 g/m2 day” should be “14.6 – 21.0 g/m2 day”, “cassava starch / gelatin films also” should be “cassava starch/gelatin films also”, and so on.
- Please standardize the format of references.
Author Response
Response to reviewers
Thank you reviewer for the good comments and suggestion. The manuscript was revised as comments. Red font and highlighting are used to indicate the answers and changes in the manuscript, respectively. The grammatical errors’d already rechecked and revised.
Reviewer #1: This research has developed new antioxidant active food packaging materials from cassava starch/gelatin (7:3 w/w) composite films varied with antioxidant types [quercetin and tertiary butylhydroquinone (TBHQ)] and concentration (0-200 mg / 200 mL film-forming solution) and evaluated their properties. At 34% RH, increasing the concentration of quercetin affects composite films by increasing the tensile strength and reducing the elongation at break. Cassava starch/gelatin composite films integrating with quercetin and TBHQ 40 can be utilized as an active packaging which can delay oxidation in foods.
- “However, there was no effect on EAB of the cassava starch/gelatin films when quercetin concentration is increased.”. The authors should explain this result.
Answer: The whole paragraph was revised for more clear explanation as “By comparing the mechanical properties of quercetin and TBHQ films, it is evident that cassava starch/gelatin composite films containing quercetin had a higher tensile strength than TBHQ-containing films. Additionally, cassava starch/gelatin films with quercetin were less flexible than films with TBHQ. The composite films with highest TS were films made from 50 mg quercetin/200 mL film solutions and stored at 34%RH, while the highest EAB films were made from 100 mg TBHQ/200 mL film solutions and stored at 54%RH. These findings were not consistent with a report by Rachtanapun et al [26], who studied the addition of antioxidants (PG, BHA, BHT) to rice flour/cassava starch films and found that the type of antioxidants had no effect on the mechanical properties of the film [26]. In terms of the storage RH effects on the film mechanical properties, the EAB of films stored at 54 %RH was greater than those stored at 34 %RH, but the TS were lower. Increasing storage RH reduced intermolecular interactions and increased polymer mobility, both of which result from water acting as a plasticizer by binding with hydroxyl groups (OH) in the starch chain [39,40]. Others have reported similar RH-dependent effects on film mechanical properties [41].”
- 3 (XRD) 8 (a). are not very clear. Please improve resolution of the Figs, especially the numbers and words.
Answer: XRD figure was adjusted as comment.
- “In addition, a strong water band at 1407 cm-1, associated with OH in-plane bending, is discernible in the films with antioxidants.”. One reference is needed.
Answer: Ref [33] was added.
- FTIR spectra are recommended to be placed in the text.
Answer: FTIR spectra was added as Figure 2.
- The letter notation of Duncan’s test using ANOVA in Table 2 was all wrong. Please correct the rule based on letter marking.
Answer: All letter notation was corrected.
- “3.5 Film morphology”. The author should discuss the differences between the treatments and explain the potential significance.
Answer: The SEM images of all films showed almost the same, so the paragraph was slightly modified for more clear explanation as “The SEM image of cassava starch/gelatin film showed a dense structure with a few areas of roughness (Figure 4a), that can happen due to the versatility of gelatin films [52]), which proved a homogeneous structure. The micrographs of cassava starch/gelatin films with quercetin and TBHQ (Figure 4b and 4c, respectively) exhibited a dense and rough cross-section. More cracking was seen in the SEM micrograph of the TBHQ-containing films, which might have been caused by phase separation or inhomogeneity of TBHQ in the polymer matrix and could have contributed to the lower tensile strenghts of the TBHQ-containing films (Figure 1)”.
- Combine Figures 2a and 2b into one picture. Figure 3 is the same process.
Answer: Figure 2 and 3 were adjusted as comments and rearranged as Figure 3 and 4.
- Many irregularities were described in the manuscript. For example, “14.6 – 21.0 g/m2 day” should be “14.6 – 21.0 g/m2 day”, “cassava starch / gelatin films also” should be “cassava starch/gelatin films also”, and so on.
Answer: All was edited.
- Please standardize the format of references.
Answer: Reference format already revised.

Reviewer 2 Report
This mamuscript intitled "Antioxidant Films from Cassava Starch/Gelatin Biocomposite Fortified with Quercetin and TBHQ and Their Applications in Food Models" is well presented and organized. It could be published after the following revisions:
- Abstract: "The edible and active packaging is widely used as food packaging due to their eco-friendliness and safety consumable." I don't think that in this time edible and active packaging films are widely used. Although I believe that are going to widely used in the future...So rephrase this sentence pls.
- Introduction lines 101-106. Separate a paragraph and specify better the objectives of this manuscript.
- 2.2 Film preparation. Pls provide a table with code names, %content and amounts of starch/gelatin/quercetin and TBHQ used for the preparation of each film.
- lines 258-260: "The cassava starch/gelatin film with quercetin showed higher TS than the control at 34 %RH because hydroxyl groups of quercetin could probably act as hydrogen donors, and formed hydrogen bonding with starch/gelatin molecules" Add a reference for this result please. Find similar results for quecetin added in starch or other polymers biopolymers.
- line 262: "On the other hands, films containing TBHQ has reported to have low TS" it seems that a reference missing here...
- In the SEM images put in the uper right corner images of each film. It would be halpfull for readers to see the real image of such films and combine it with the SEM morphology.
Author Response
Response to reviewers
Thank you, reviewer, for the good comments and suggestion. The manuscript was revised as comments. Red font and highlighting are used to indicate the answers and changes in the manuscript, respectively. The grammatical errors already rechecked and revised.
Reviewer #2: This mamuscript intitled "Antioxidant Films from Cassava Starch/Gelatin Biocomposite Fortified with Quercetin and TBHQ and Their Applications in Food Models" is well presented and organized. It could be published after the following revisions:
- Abstract: "The edible and active packaging is widely used as food packaging due to their eco-friendliness and safety consumable." I don't think that in this time edible and active packaging films are widely used. Although I believe that are going to widely used in the future...So rephrase this sentence pls.
Answer: The sentence was changed to “Edible and active packaging are attractive for use in food packaging applications due to their functionality and sustainability”.
- Introduction lines 101-106. Separate a paragraph and specify better the objectives of this manuscript.
Answer: Paragraph leading to the objectives was separated and revised some sentences.
- 2.2 Film preparation. Pls provide a table with code names, %content and amounts of starch/gelatin/quercetin and TBHQ used for the preparation of each film.
Answer: Table M1 was added for film composition.
- lines 258-260: "The cassava starch/gelatin film with quercetin showed higher TS than the control at 34 %RH because hydroxyl groups of quercetin could probably act as hydrogen donors, and formed hydrogen bonding with starch/gelatin molecules" Add a reference for this result please. Find similar results for quecetin added in starch or other polymers biopolymers.
Answer: Ref. [41] was added.
- line 262: "On the other hands, films containing TBHQ has reported to have low TS" it seems that a reference missing here...
Answer: The sentence was revised as “Increasing quercetin concentration to 200 mg and increasing storage RH both decreased TS of the films compared to the control. In contrast, films containing TBHQ had lower TS than the control at all TBHQ concentrations, attributed to possible incompatibility of the TBHQ and the starch/gelatin polymers which potentially disrupted the polymer network thereby decreasing film strength. This result is consistent with reports of TS of fish skin gelatin film incorporated with BHT and α-tocopherol [34]”.
- In the SEM images put in the upper right corner images of each film. It would be helpful for readers to see the real image of such films and combine it with the SEM morphology.
Answer: The alphabets (a, b, c) were added in the upper right corner of each SEM image.

Round 2
Reviewer 1 Report
Based on the author's revision, this article can be accepted.Reviewer 2 Report
Author's replies to all of my comments.
I suggest publication as it is.
Best Wishes